physical chemistry

corrosion, stainless steel, ionic liquids, tris(pentafluoroethyl)trifluorophosphate, surface

**Author for correspondence:**
Imee Su Martinez
e-mail: ismartinez@up.edu.ph

This article has been edited by the Royal Society of Chemistry, including the commissioning, peer review process and editorial aspects up to the point of acceptance.

# Exploring the corrosion inhibition capability of FAP-based ionic liquids on stainless steel

Julius Kim A. Tiongson[1], Kim Christopher C. Aganda[4], Dwight Angelo V. Bruzon[2], Albert P. Guevara[4], Blessie A. Basilia[4], Giovanni A. Tapang[2] and Imee Su Martinez[1,3]

[1]Natural Sciences Research Institute, [2]National Institute of Physics, and [3]Institute of Chemistry, University of the Philippines Diliman, Quezon City 1101, Philippines
[4]Advanced Device and Materials Testing Laboratory, Department of Science and Technology Compound, Gen. Santos Avenue, Bicutan, Taguig City 1631, Philippines

ISM, 0000-0001-9512-4089

Corrosion is clearly one of the more common causes of materials failure in stainless steel. To manage corrosion, chemical inhibitors are often used for prevention and control. Ionic liquids due to their hydrophobic and corrosion-resistant property are being explored as alternative protective coatings and anti-corrosion materials. In this particular study, ionic liquids containing functionalized imidazolium cations and tris(pentafluoroethyl)trifluorophosphate (FAP) anions were investigated for their ability to inhibit corrosion on stainless steel surfaces in acidic environment. Using surface characterization techniques, specifically scanning electron microscopy and energy-dispersive X-ray (EDX), the morphology and the elemental composition of the steel surfaces before and after corrosion were determined. Contact angle measurements were also performed to determine how these ionic liquids were able to wet the stainless steel surface. In addition, potentiodynamic studies were carried out to ensure that corrosion inhibition has occurred. Results show that these ionic liquids were able to inhibit corrosion on the stainless steel surfaces. This indicates promise in the use of these FAP-based ionic liquids for corrosion management in stainless steel.

## 1. Introduction

Stainless steel is widely used in various applications such as in construction, maritime transport and aerospace engineering [1,2].

Grades of stainless steel vary depending on the amount of chromium amalgamated into the iron. The presence of other components such as molybdenum, nickel, titanium and copper, as well as non-metal additives like carbon and nitrogen, also affect its classification [3]. Among the many types of stainless steel, the austenitic grade is noted for its increased corrosion resistance, good durability and relatively high tensile stress [4]. The nickel-containing types are classified as the 300 series, which are more formable and ductile compared to the manganese-containing ones, designated as the 200 series stainless steel [1,3]. Although these materials are known to be corrosion resistant, they may deteriorate when exposed to static stress and hostile environments, such as in acidic, alkaline and saline media [5–7]. The risk is even more pronounced in lubricated steel or steel contacts, and at high temperature and high pressure conditions common in industrial plants [7].

Corrosion occurs when the oxidation of a metal is coupled with a more positive reduction reaction such as with oxygen gas in acidic water [8]. Understanding this mechanism is key to prolonging the lifetime of stainless steel structures. Different forms of corrosion-protection techniques are used to reduce the rate of surface corrosion in order to extend the lifetime of stainless steel. These include removal of the oxidizing agents such as boiler water treatment, prevention of surface reaction such as employment of sacrificial anode or impressed current, and inhibition of surface reaction by applying chemical inhibitors. Among these corrosion inhibition methods, the use of chemical adsorbents or coatings has been the most practical solution in different applications [8–12]. The effectiveness of using organic coatings on metals against corrosion from acidic environment is in fact established. The more common ones such as paints, claddings and surfactants are considered mature technologies in anti-corrosion applications. However, research into the development of new compounds can still provide significant contribution in terms of improved corrosion inhibition properties, enhanced adhesion properties, lower cost and lengthier lifetime of applied material.

Ionic liquids are currently being developed as alternative anti-corrosion coating materials [13–16]. They are composed of organic salts in melted form at temperatures below 100°C [17,18]. They are considered green materials due to their negligible vapour pressure. They also have other interesting properties such as high degradation temperature, chemical stability, non-flammability and high heat capacity, which are advantageous for this type of application. These liquids are also tuneable, which means that ions can be functionalized in order to have specific properties, or the combination of anions and cations may be varied to come up with an ionic liquid with a specific application in mind [17,19,20]. In fact, these liquids can be tailored to have high activity in acidic medium [21,22]. An amphiphilic configuration, for example, with a long-chain hydrophobic tail and hydrophilic polar head, allows micelle formation, which may inhibit corrosion by reducing the ability of acidic media to penetrate into the steel [23].

In most studies involving ionic liquids as anti-corrosion materials, the imidazolium cation is used due to its propensity to adsorb on the metal surface [24–27]. In addition to its electronegative nitrogen and –C=N– bond, tethering some functional groups may create additional interaction between the ionic liquid and the metal surface enhancing corrosion inhibition in metals [25–28]. A recent study by Cao *et al.* [29], which involved both theoretical and experimental work, has shown the effectiveness of a diimidazolium ionic liquid with phenyl and –COOH functional moieties as mixed-type corrosion inhibitor in the presence of potassium iodide. The cations arranged themselves parallel to the surface leading to increased coverage and surface protection. Another study has also shown the effectiveness of imidazolium-based ionic liquids as corrosion inhibitors for stainless steel in acidic environment [30]. The fluorine-based anion $BF_4^-$ was observed to have a key function in the adsorption of the ionic liquids on the steel surfaces. The hydrophobic nature of the ionic liquid is also important for corrosion inhibition, as shown in the work of Herrmann *et al.* [31]. They were able to develop hydrophobic polyoxometalate ionic liquids that were capable of preventing corrosion in copper in the presence of acetic acid vapours, and simulated acid rain.

The tris(pentafluoroethyl)trifluorophosphate (FAP) anion is gaining special attention due to its excellent hydrophobic property brought about by its large non-polar perfluoroethyl groups [32–34]. FAP-based ionic liquids are considered as replacements for ionic liquids that are hydrolytically unstable, especially in aqueous media [32,35]. In addition, these types are thermally and electrochemically stable, making them ideal for anti-corrosion application [34]. In fact, a previous study performed by our group was able to show that properties of FAP ionic liquids are comparable to that of the well-studied and well-performing $Tf_2N$-based ionic liquids [36]. Moreover, these FAP ionic liquids were determined to possess even better electrochemical stability showing wider potential windows, keeping in mind of course counter cations and differences in electrodes used. To the best of our knowledge, there are no reported studies yet on the corrosion inhibition of FAP-based ionic liquids on stainless steels in acidic medium. The aim of this work is to investigate the anti-corrosion capabilities of ionic liquids containing functionalized imidazolium cations and FAP anion on austenitic stainless steel in acidic environment. Surface characterization techniques particularly

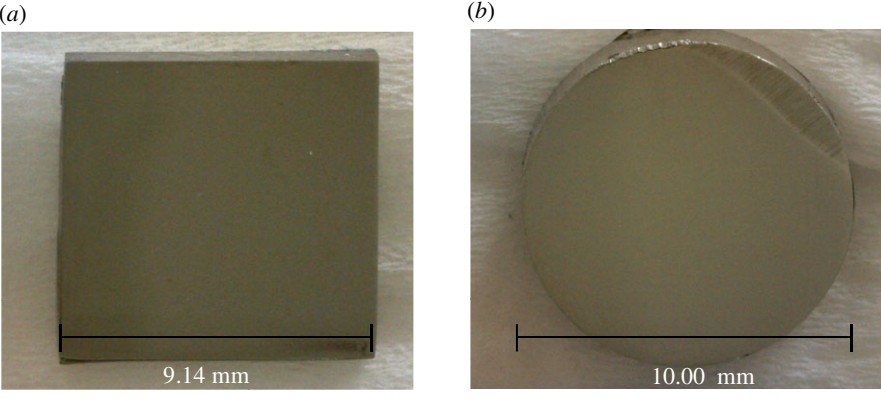

**Figure 1.** Optical images of the polished S30400 (*a*) and S20200 (*b*) samples. Captured at 20× magnification under an optical microscope.

**Table 1.** Ions comprising the ionic liquids under study.

| chemical name | abbreviation | structure |
|---|---|---|
| 1-pentyl-3-methylimidazolium | [PMIM] | |
| 1-(4-methoxybutyl)-3-methylimidazolium | [MOBMIM] | |
| tris(pentafluoroethyl)trifluorophosphate | [FAP] | |

scanning electron microscopy (SEM), energy-dispersive X-rays (EDX), contact angle (CA) measurements and potentiodynamic polarization scanning were used to test these ionic liquids.

# 2. Material and methods

## 2.1. Preparation of FAP-based ionic liquids

The ionic liquids used in this study have a common anion, and imidazolium cations with two different functional groups at the terminus of the alkyl chain. Table 1 shows the FAP-based ionic liquids used in this study. The ionic liquids were prepared according to a previously reported method by our group [36]. The spectroscopic analyses, water content and chloride impurities of these ionic liquids were also presented in the same study, showing the purity of these samples. All of the FAP-based ionic liquids were pre-evacuated to around $5.0 \times 10^{-5}$ torr using a high vacuum line to remove residual volatile organic impurities, before they were applied on the stainless steel substrates.

## 2.2. Preparation of the stainless steel substrates

A Topper precision cutting machine (CL-50) was used to cut flat bars of stainless steel type 304 (S30400) into rectangular pieces of dimensions $9.14 \times 9.36 \times 0.20$ mm. Cylindrical rods of stainless steel type 202 (S20200) were also cut into cross-sections of diameter 10.00 mm and thickness of 5.00 mm. The substrates were

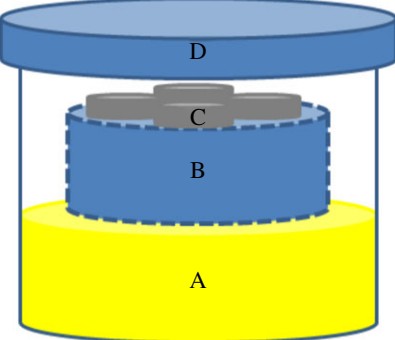

**Figure 2.** Acid corrosion set-up made up of a glass container. (A) Fifty per cent aqueous formic acid solution; (B) elevated Teflon mount; (C) container for the stainless steel substrates; (D) lid.

polished into a mirror finish using a Topper grinding and polishing machine (P20FS-1-A3-HA, P20F-I-HA), a rotating pan and increasing grades of silicon carbide sandpapers. The stainless steel substrates were cleaned by ultrasonication for 5 min with deionized water. Optical images of the processed stainless steels were captured using a Dinolite AM-4013 with a magnification of 20× as shown in figure 1.

## 2.3. Contact angle measurements on stainless steel substrates

Contact angles (CAs) of the FAP-based ionic liquids (ILs) and deionized water on the prepared stainless steel substrates were measured at ambient conditions. A home-built surface tensiometer was used to take the images of the drops every minute for a duration of 10 min. DropSnake ADSA software from ImageJ was used to determine the CAs of the drops.

## 2.4. Application of the FAP-based ionic liquids on the stainless steel substrates

This was performed according to the method used by Herrmann *et al.* [31]. The stainless steel substrates were cleaned first through sonication for 10 s in deionized water, rinsed with acetone and then dried with nitrogen gas. Approximately 50 µl of the FAP-based ionic liquids was applied to the stainless steel substrates through drop-coating. The freshly coated substrates were immediately stored in a dessicator before analyses.

## 2.5. Acid exposure of the stainless steel surface

The acid exposure method used in this study is based on the works of Likhanova *et al.* and Herrmann *et al.* [24,31]. Figure 2 shows the acid corrosion set-up containing 50% formic acid. Acid vapour was allowed to come in contact with the substrates, instead of directly immersing the substrates into the solution. This is to simulate hostile environments that stainless steel structures are subjected to, as described in previous studies [24,31]. Ionic liquid-free stainless steel substrates were also subjected to the same set-up to serve as reference.

## 2.6. Scanning electron microscopy and energy-dispersive X-ray spectroscopy

The ionic liquid-free and ionic liquid-coated stainless steel substrates before and after acid exposure were analysed through SEM and EDX. The coated substrates after acid exposure were rid of ionic liquids through rinsing with water and acetone, and then dried using $N_2$ gas before they were analysed. A Dual Beam Helios Nanolab 600i SEM with a 3 kV voltage in the backscattering mode (BSE) was used.

## 2.7. Potentiodynamic studies

The electrochemically active surface area of the fabricated stainless steel electrode was determined through steady-state voltammetry. A 2.0 mM Ferrocene solution in acetonitrile was used with a 0.1 M tetrabutylammonium perchlorate supporting electrolyte. The electrode was covered with a printed housing fabricated using an Ultimaker 2+ printer, and a thermoplastic polyurethane flexible filament to maintain a fixed area of exposed stainless steel in solution.

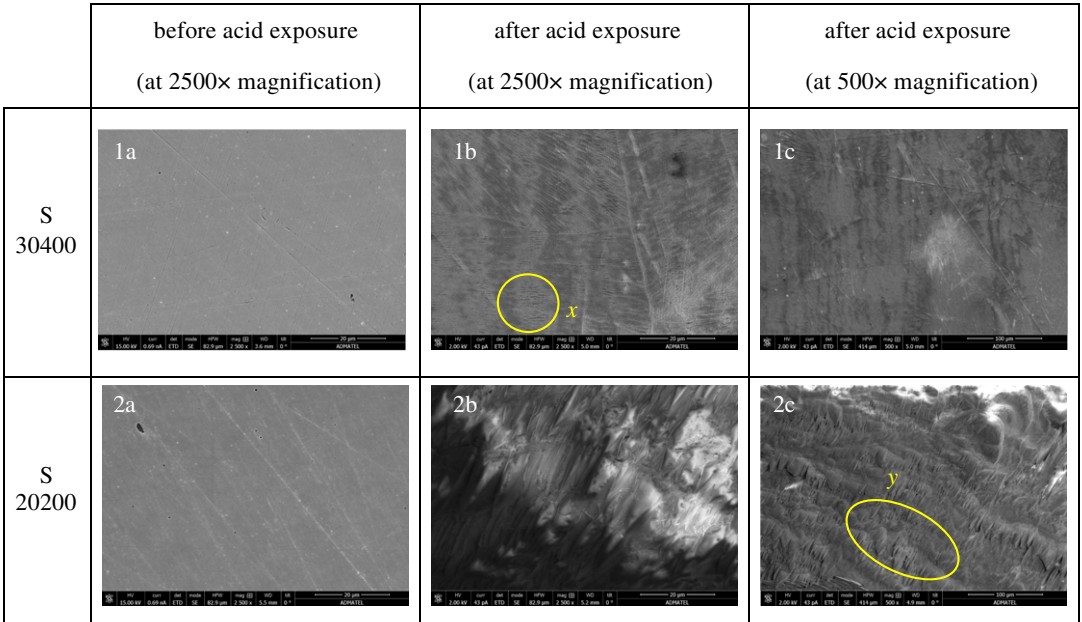

**Figure 3.** SEM of uncoated stainless steel substrates. S30400 surface: (1a) before acid exposure at 2500× magnification, (1b) after acid exposure at 2500× magnification, (1c) after acid exposure at 500× magnification. S20200 surface: (2a) before acid exposure at 2500× magnification, (2b) after acid exposure at 2500× magnification, (2c) after acid exposure at 500× magnification.

Potentiodynamic experiments were performed on solutions containing varying amounts of [MOBMIM][FAP] dissolved in 1.0 M HCl. Measurements were conducted using an EDAQ e-corder 410 potentiostat employing a three-electrode system. Linear staircase voltage ramps with a scan rate of $0.5\,\text{mV s}^{-1}$ were applied in the anodic and cathodic directions upon stabilization of the open circuit potential. The step height was at $-1$ mV, while the step width was 2000 ms. Measurements reported in this work are average values.

The inhibition efficiency (IE) was calculated from the extrapolated values of the corrosion exchange current density using the equation [37]

$$\text{IE}\ (\%) = \frac{J_{\text{corr}}^{\text{o}} - J_{\text{corr}}}{J_{\text{corr}}^{\text{o}}} \times 100\,,$$

where $J_{\text{corr}}^{\text{o}}$ is the corrosion exchange current density in the absence of an inhibitor and $J_{\text{corr}}$ is the corrosion exchange current density in the presence of an inhibitor.

# 3. Results and discussion

## 3.1. SEM–EDX corrosion studies on ionic liquid-free stainless steel substrates

SEM images of the ionic liquid-free or uncoated stainless steel substrates were acquired before acid exposure to ensure that initial microstructure deformation on the stainless steel surfaces, which can be mistaken for corrosion, are not present. Images 1a and 2a in figure 3 show that smooth surfaces were observed on both grades of stainless steel. Except for evenly spaced diagonal markings formed during the polishing process, cracks and other deformations were not observed on the sample surface.

The acid vapour exposure time was optimized for both samples. Seventy-two hours were necessary to corrode the S30400 substrate using vapour coming from 50% formic acid, while 24 h was the needed exposure time for the S20200. Both of the surface morphologies of the corroded stainless steels displayed porous marks and cracks shown in figure 3, images 1b, 1c, 2b and 2c. Even at lower magnification of 500×, corrosion on both surfaces was clearly exhibited. In the corroded surface of S30400 at 500× magnification, it can be observed that general corrosion from the acid has appeared leading to a rougher morphology on the surface. A closer inspection at 2500× shows that striations were formed on the surface of the metal, emphasized by the circular perimeter $x$. The dimensions of these morphological deformations are 2.59–5.92 μm in length and widths of 0.35–0.47 μm. These striations are probably localized crevice corrosion that occurred at the longitudinal polishing marks. Another

**Table 2.** Chemical composition of the tested stainless steel.

| | % of element | | | | | | |
|---|---|---|---|---|---|---|---|
| | C | Fe | Mn | Cr | Si | Ni | PREN[a] |
| TYPE | | | | | | | |
| S30400[b] | 0.08 | bal | 2.0 | 18.0–20.0 | 1.00 | 8.0–10.5 | 18.0–20.0 |
| S30400[c] | 4.0 ± 0.2 | 69.1 ± 0.1 | — | 18.8 ± 0.1 | 0.5 ± 0 | 7.6 ± 0.1 | 18.8[d] |
| TYPE | | | | | | | |
| S20200[b] | 0.15 | bal | 7.5–10 | 17.0–19.0 | 1.00 | 4.0–6.0 | 17.0–19.0 |
| S20200[c] | 6.4 ± 0.2 | 72.0 ± 0.5 | 10.8 ± 0.2 | 8.5 ± 0.1 | 0.3 ± 0.1 | 1.3 ± 0.1 | 8.5[d] |

[a]Pitting resistance equivalence number.
[b][38].
[c]Experimental results.
[d]Calculated from experimental EDX results.

possibility is the presence of filiform corrosion knowing that the environment is highly humid and acidic. Longitudinal protuberances with width size ranges of 25.54–29.47 µm were observed on the S20200 surface leading to a frosted appearance. At both magnifications, the change in surface morphology in the presence of the acid is quite apparent. The corrosion, as can be seen in figure 3, images 2b and 1b, is much more pronounced in S20200 than on S30400. Surface deformations on the S20200 at 500× magnification are emphasized by an oblate perimeter $y$.

The elemental compositions of these stainless steel grades are known. Although S30400 is expected to be more resistant to corrosion than S20200, based on the pitting resistance equivalent value or number (PREN), the difference is slight, and the amount of chromium in these types are comparable. The presence of higher amounts of Mn in S20200 is also expected to assist in corrosion prevention by providing a passive layer in acidic environment, just like Ni. However, in the experimental results, the rate at which the surface of S20200 corroded was observed to be three times faster than that of S30400, based on the needed exposure time for corrosion to occur. In terms of the SEM images, the degree of corrosion on S20200 is also more pronounced. Table 2 shows the compositions of both stainless steel types from the literature and from experimental results. From these data, it can be seen that the tested stainless steel deviated from standard values in terms of content. The composition, particularly S20200, is quite far from the expected amounts of chromium in the substrate. The presence of high levels of carbon and low levels of chromium may have contributed to its fast corrosion rate [39]. It is possible that the passive layer of chromium carbide did not form, given that the composition of chromium is below 10.5%, and the carbon levels are relatively high. Another possibility is that the low amount of chromium lessened the amount of $Cr(OH)_3$ hydrophobic layer from forming leading to an enhanced propensity towards corrosion [40,41]. Low chromium content led to a low PREN making the S20200 surface more susceptible to corrosion. For both samples, the determined amounts of chromium from the EDX results correlate with the observed degree of morphological deformations due to corrosion, as well as propensity towards corrosion.

Figure 4 shows changes in the chemical composition of the substrates before and after exposure to formic acid. The increase in oxygen content and decrease in amounts of iron are used to indicate the presence of corrosion. Drastic increase in oxygen content and decrease in iron content was more pronounced in S20200. A change from 0 to 50% for oxygen, and a decrease in iron content from 70 to 30% was observed. In fact, oxygen is the highest elemental presence after corrosion in S20200. This shows the expected predisposition of S20200 to corrode, given the initial EDX data of very low chromium content. The amounts of chromium and manganese in S20200 also decreased after exposure to the acid vapour.

For S30400, the initial EDX results prior to acid exposure were at least close to standard value. This correlates with the experimental results showing that SS30400 is more resistant to corrosion. The appearance of manganese in the corroded S30400 is probably due to the presence of manganese in the lower stratum of the sample. Nickel and chromium were also observed to decrease in amounts in the S30400 substrate. The tabulated data of the EDX results are shown in the electronic supplementary material (SI–SIII). The oxides present in S30400 and S20200 were probably FeO, $Fe_2O_3$ and $Fe_3O_4$ due to large amounts of iron and oxygen in both samples [7].

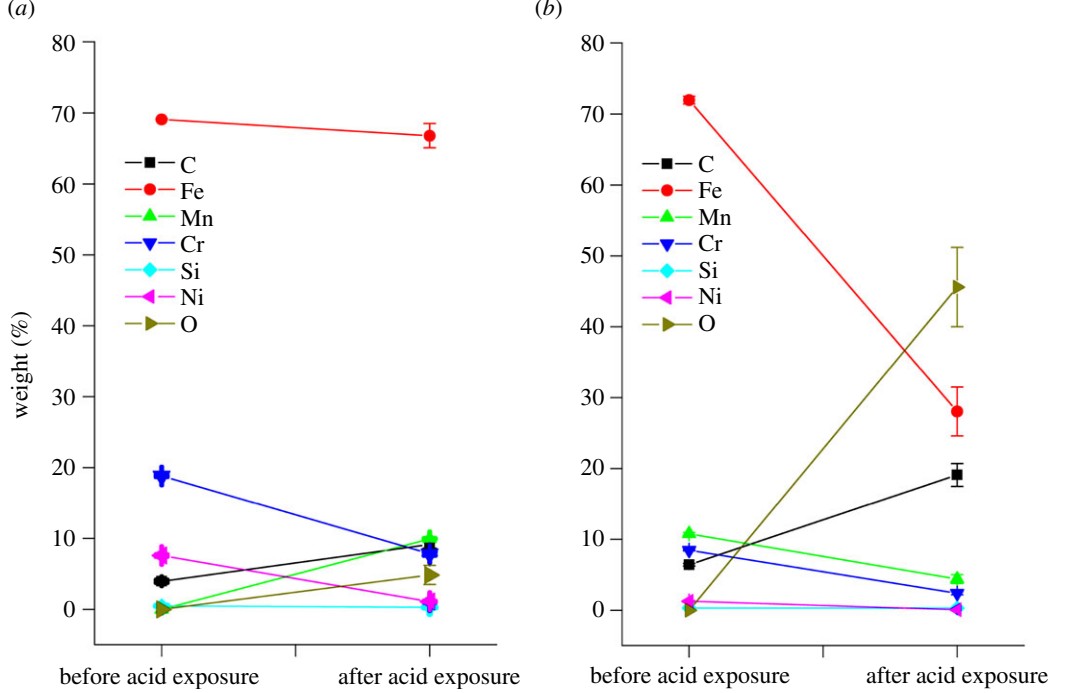

**Figure 4.** Chemical compositions of (*a*) S30400 and (*b*) S20200 substrates from EDX results.

## 3.2. Contact angle measurements on clean stainless steel substrates

The measured CA values of water on the stainless steel surfaces show that the substrates used for the experiment are clean and freshly polished [42]. The presence of organic contaminants, for example, on the solid surface may lead to CAs greater than 90°.

CA measurements were performed to determine the affinity of the ionic liquids to the stainless steel, which can affect their corrosion inhibition property. The FAP ionic liquids tested for this study were observed to wet the surfaces of the stainless steel substrates. This is expected as stainless steel substrates have high surface energies [3]. The CAs of the FAP-based ionic liquids and water on S30400 and S20200 substrates were plotted against time, as shown in figure 5. These values are listed in detail in the electronic supplementary material. A decrease in the measured CA was observed through time, indicating spreading of the liquid on the surface. S20200 was observed to have relatively lower CAs compared with that of S30400 indicating a higher surface energy, and higher affinity towards the ionic liquids.

Based on the results, the FAP-based ionic liquids were more capable of wetting stainless steel than water. This may be attributed to the nature of the stainless steel surfaces. Since metal atoms are very electron deficient due to metallic bonding, their electron densities are reduced, which gives them higher affinity towards electron-rich substances [43]. In fact, previous studies have shown the adsorption of ionic liquid ions to polar substrates depending on the predominant excess charge on the interface [44].

According to Pereira and co-workers [43,45], basic liquids may adhere more to metallic surfaces due to intermolecular forces of attraction. Given that fluorinated anions have higher hydrogen-bond basicity, the ionic liquids containing FAP anions are expected to easily wet metal surfaces. For the cation, the alkyl and methoxy group of the [PMIM][FAP] and [MOBMIM][FAP] are electron-donating leading to enhanced basicity.

Results show that [PMIM][FAP] gave a lower CA compared to [MOBMIM][FAP] on S30400. This is probably due to the presence of more chromium in S30400, which may create a layer of hydrophobic $Cr(OH)_3$ on the stainless steel surface, enhancing its affinity towards the methyl-containing ionic liquid [40]. The opposite is true for S20200, wherein the final measured CA of [MOBMIM][FAP] is $22.0 \pm 0.3$, which is lower than the CA of [PMIM][FAP] at $24.1 \pm 0.4$. In fact, [MOBMIM][FAP] on S20200 gave the lowest measured CA, indicating enhanced wetting on the surface. The results here are congruent with the measured CAs of water indicating a more hydrophilic S20200 compared to S30400. Although the anion is known to play a dominant role in the wetting of ionic liquids unto the

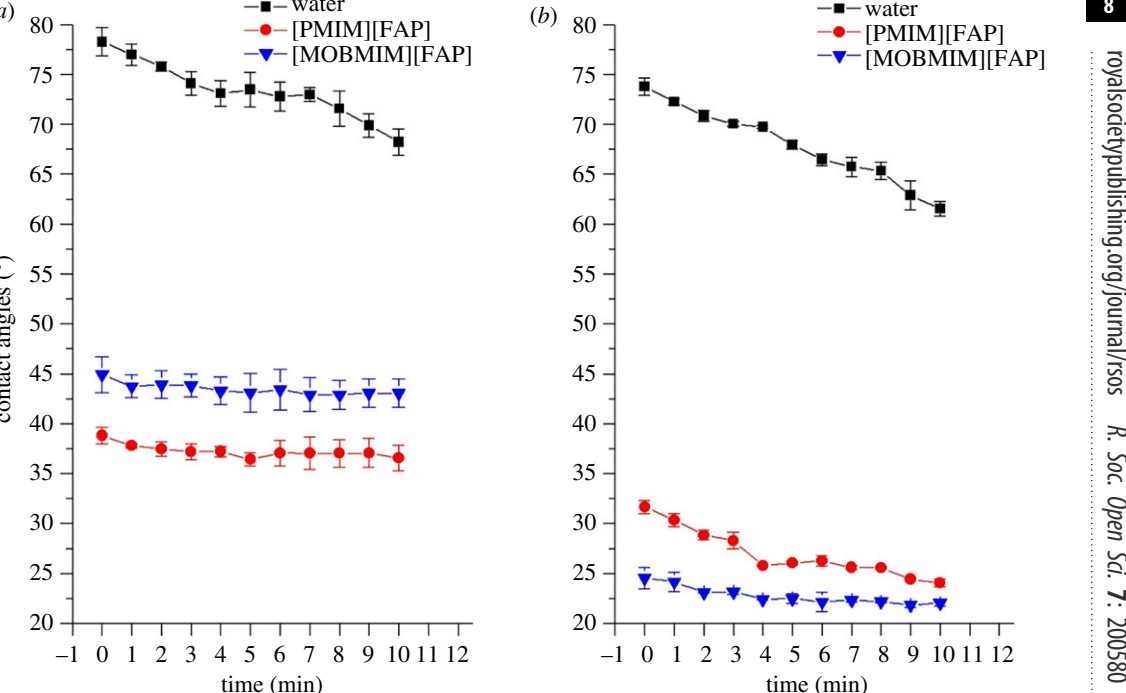

**Figure 5.** Contact angle measurement of water and FAP-based ILs on (*a*) S30400 and (*b*) S20200. Square, water; circle, [PMIM][FAP]; inverted triangle, [MOBMIM][FAP].

substrate, the cation may fairly contribute, given the influence of the different functional groups attached to it, such as the tethered methoxy moeity [43].

## 3.3. SEM–EDX of the FAP ionic liquid-coated stainless steel substrates

SEM images were taken of the S30400 samples coated with ionic liquids. Figure 6 shows these images at 500×. The regions indicated by a are the areas covered by the ionic liquids, whereas regions b are the areas without the ionic liquids. After acid exposure, the ionic liquids formed tiny spherical globules on the stainless steel surfaces. The diameters of these circular features are 0.03–0.13 μm in size for the [PMIM][FAP], and more significant for the [MOBMIM][FAP] at diameters of approximately 4 μm. The formation of these micrometre-sized globules in the presence of the aqueous acid is probably due to excess positive charge on the stainless steel leading to aggregation or micelle formation. Ionic liquids are actually known to aggregate and form micelles in aqueous solutions [46]. The aqueous acid having a pH lower than the isoelectric point of the S30400 stainless steel caused its surface to be covered predominantly with –$OH_2^+$. This must have caused the anions in the ionic liquid to situate closer to the stainless steel, and the cations to situate away or locate within the confines of the anions. This is perhaps the same reason why these microglobules disappear and spread during high-voltage exposure, where there is a stream of electrons bombarding the surface. The excess negative charge attracts the cations of the ionic liquid causing the droplets to spread on the stainless steel surface. The fact that the ionic liquids are hydrophobic may have also caused this observed globule formation. Figure 7 shows the effect of the acid and high-voltage exposure to the ionic liquids on the stainless steel. Further studies, however, are necessary to understand this behaviour. This may have some interesting implications on ionic liquid charge arrangement at electrified surfaces.

Complementary to the morphological data, figure 8*a,b* shows the EDX results for both ionic liquids on the S30400. All of the expected elemental contents of [PMIM][FAP] and [MOBMIM][FAP] were clearly present on the S30400, except nitrogen. Nitrogen was not detected due to the low count rate of EDX analysis for light elements [47]. Even with the ionic liquid coating, the chemical components of S30400 such as iron, chromium, manganese, carbon, silicon and nickel were also detected. The amount, however, was reduced in comparison with the bare steel substrate. After acid exposure of the ionic liquid-coated S30400, there was not much change in the elemental composition of the

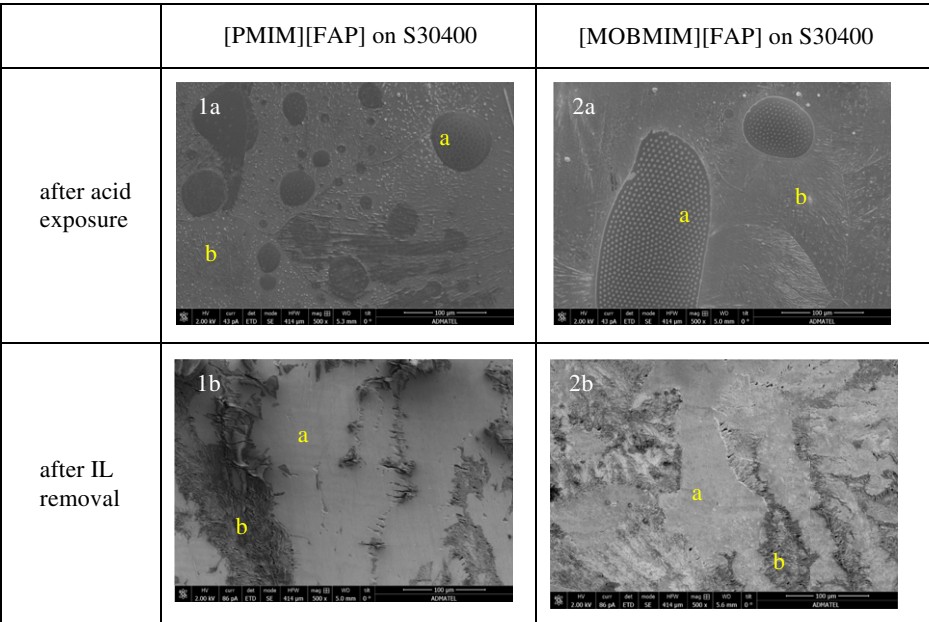

| | [PMIM][FAP] on S30400 | [MOBMIM][FAP] on S30400 |
|---|---|---|
| after acid exposure | 1a | 2a |
| after IL removal | 1b | 2b |

**Figure 6.** Ionic liquid on S30400.

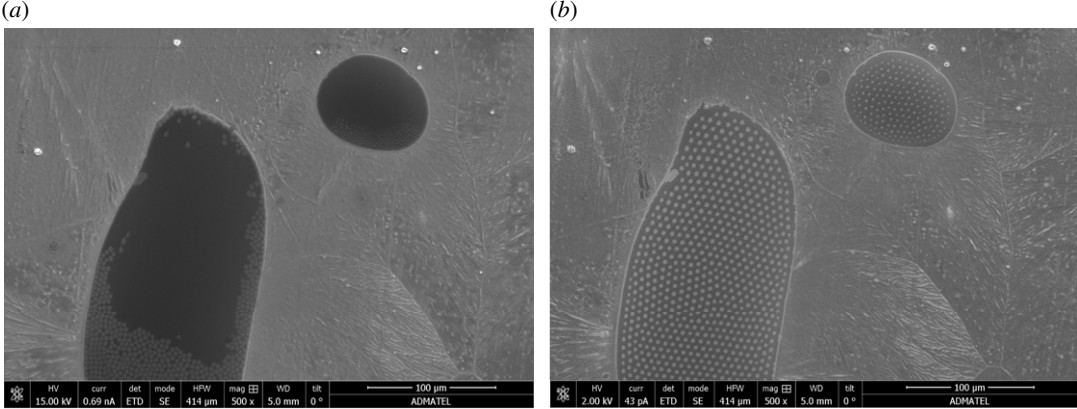

**Figure 7.** SEM image of acid-exposed [MOBMIM][FAP]-coated S30400 after taking EDX data (*a*) and prior to taking EDX data (*b*). Captured at 500× magnification.

samples, as can be seem as you go from the blue columns to the red columns in figure 8. Nickel was seen to increase after acid exposure probably due to the interaction of formic acid with the first layer of the stainless steel, leading to the exposure of nickel to the surface. Washing the metal substrates after acid exposure to remove excess or remaining ionic liquids shows significant elemental changes to the stainless steel surface. Percentages of elements coming from the ionic liquids such as carbon and fluorine decreased significantly. The iron content of the washed stainless steel is once again at high amounts comparable to the values prior to exposure. The oxygen content was also similar to the uncoated samples showing the corrosion resistance of S30400. In terms of the effectiveness of the ionic liquid as corrosion resistance material based on the amounts of oxygen, this cannot be determined from the results for S30400. The SEM images, however, show smoother morphologies where the ionic liquids were once situated. Figure 6, images 1b and 2b, shows how the surfaces looked after removal of the ionic liquid. The elongated lumps and dents that are very abundant in the positive control were lessened.

In S20200, corrosion inhibition in the presence of the ionic liquids is more pronounced. [MOBMIM][FAP] was able to significantly inhibit corrosion as shown in image 2a of figure 9. The difference in appearance of the ionic liquid-coated area designated as region a, and the ionic

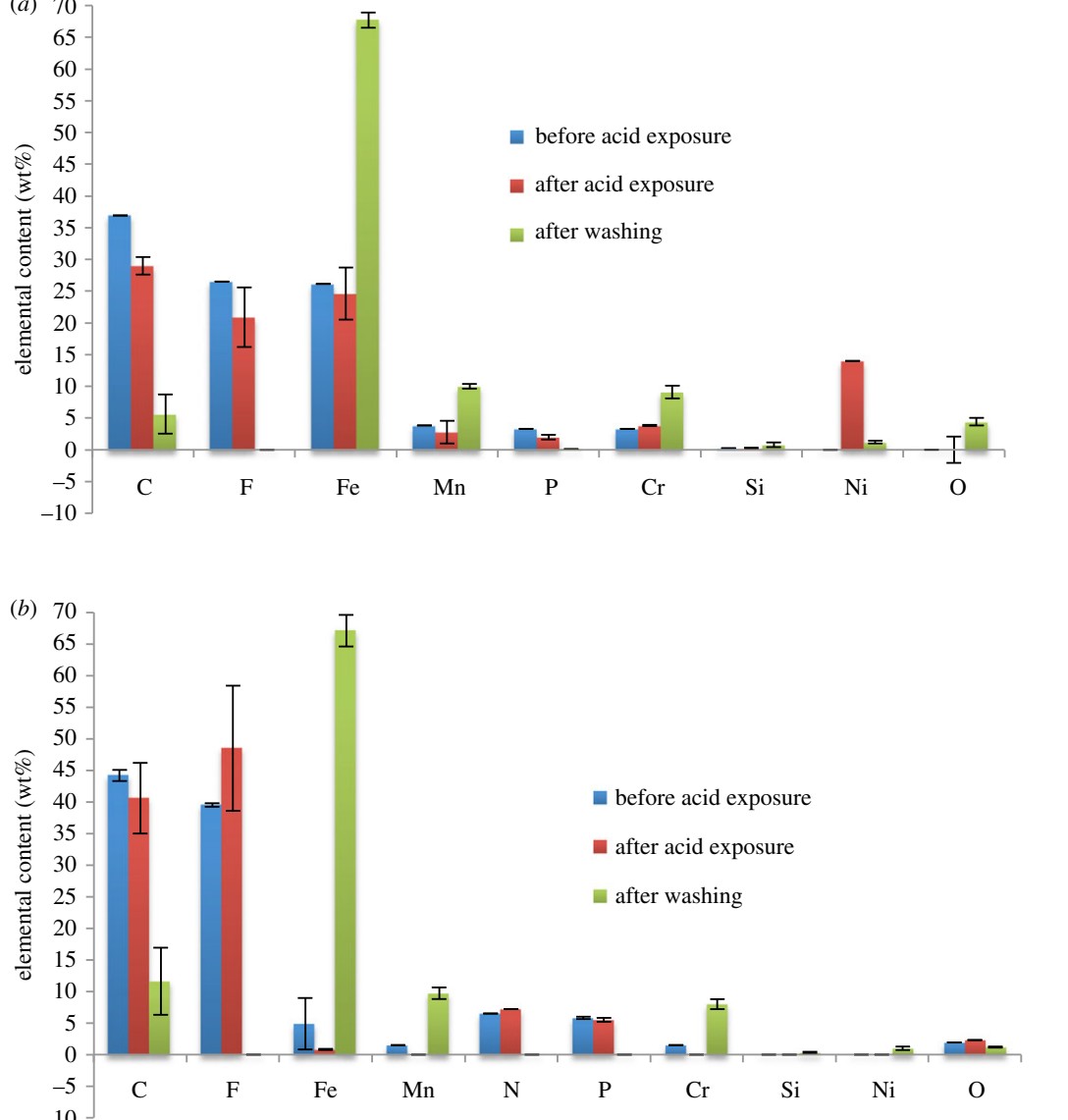

**Figure 8.** EDX results on the S30400 stainless steel substrate. (*a*) S30400 with [PMIM][FAP] and (*b*) S30400 with [MOBMIM][FAP].

liquid-free area labelled as region b is quite significant. In the area around the ionic liquid (region b), the surface evidently showed deformation, which appeared as scaling on the surface, while that of region b exhibits a smooth surface. Image 1a also shows how the presence of [PMIM][FAP] was able to protect the surface. After washing the S20200 substrate, surfaces previously coated with ionic liquids (region a) appeared to be smoother compared to the ionic liquid-free areas (region b) as seen in images 1b and 2b of figure 9.

EDX results of ionic liquid-coated S20200 are shown in figure 10. Before acid exposure, elements pertaining to the presence of the ionic liquids such as C, F and P are clearly observed. These elements were still present after acid exposure, indicating that the ionic liquids remained on the stainless steel surface. Washing off the ionic liquids shows that the amount of iron present is similar to that of the substrate before corrosion. This shows that the ionic liquids were able to actually protect the surface of the S20200 substrate. In comparison, figure 4 with the ionic liquid-free S20200 surface undergoing corrosion shows the obvious drop in iron content and increases in oxygen content. Clearly, the presence of the ionic liquids was able to protect the stainless steel surfaces from corrosion. Complex formation in the ionic liquid coating possibly developed to inhibit corrosion in the presence of the acid. Adsorbed layers of the complexes may have been produced in the cathodic and anodic sites of the metal stabilized by electrostatic forces, or van der Waals interactions in the case of an alkyl chain [48–50].

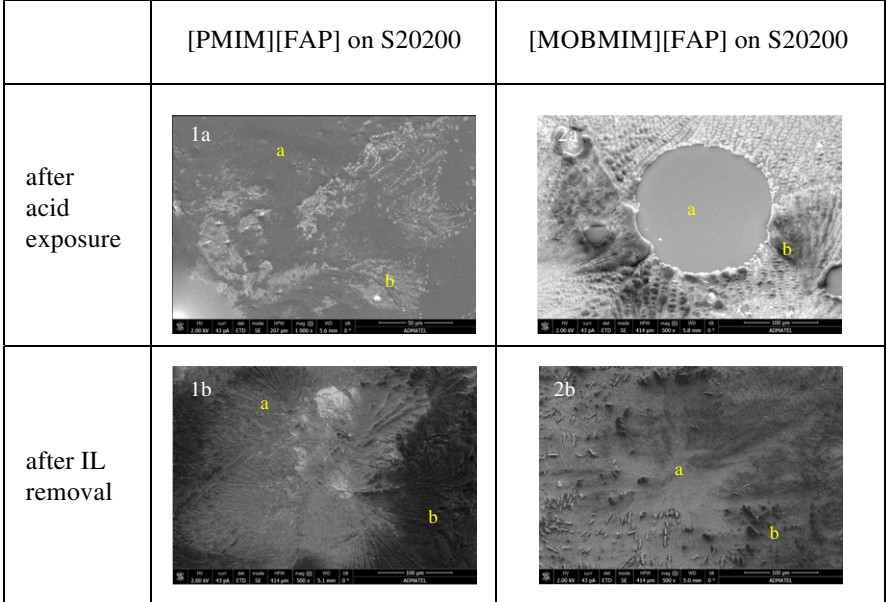

| | [PMIM][FAP] on S20200 | [MOBMIM][FAP] on S20200 |
|---|---|---|
| after acid exposure | | |
| after IL removal | | |

**Figure 9.** Ionic liquid on S20200.

The ratio of the percentage oxygen to iron content is presented in table 3. It shows how both ionic liquids are able to protect the S20200 substrate from corrosion. A ratio of about 1.62 and 0.6–0.7 was calculated for the substrate without the ionic liquid, and with the ionic liquid, respectively. S30400, which is already quite corrosion resistant, did not show significant changes in the ratio of percentage oxygen to iron. A slight change from around 0.07 without the ionic liquid to 0.02 with the ionic liquid was observed. What is also interesting about these values is the ratio 1.62 of %O to %Fe for the S20200 substrate that has undergone corrosion in the absence of the ionic liquids. This can be related to the formation of $FeO_6$ octahedral units, which are known corrosion products, previously detected using extended X-ray absorption fine structure (EXAFS) [51]. These tetramers in aqueous media form iron oxyhydroxides such as α-FeOOH through olation or oxolation [52]. In atmospheric corrosion, oxyhydroxide precipitation is influenced by dissolved oxygen, hydrogen, cation anions and other alloying elements. Since formic acid was the corrosion agent in the study, Fe(II) Fe(III) formate analogues may have been formed explaining the slight increase in carbon content after corrosion [53].

In addition, the presence of oxygen in the ionic liquid coating may be due to the reaction of the ionic liquid and the carboxylate group of the acid, similar to the findings described in the study of Likhanova *et al.* [24]. This can also be due to corrosion that may have occurred because of the penetration of some of the acid through the ionic liquid leading to its interaction with iron [7]. Traces of phosphorus were also observed in the EDX spectra. This may imply film formation due to the degradation of FAP moiety of the ionic liquid through cleavage of the C–P bond in contact with the stainless steel [47,50].

## 3.4. Potentiodynamic studies of [MOBMIM][FAP] ionic liquid on S20200 stainless steel

SEM images have clearly shown the ability of [MOBMIM][FAP] to protect the surface of S20200. Inhibition was also confirmed through EDX, where the ratio of percentage oxygen composition to iron has considerably decreased in the presence of the [MOBMIM][FAP] ionic liquid. In order to ensure the ability of this ionic liquid to prevent corrosion of the stainless steel, potentiodynamic studies were performed. Results shown in table 4 indicate that in the presence of 0.30 mM of [MOBMIM][FAP], the percentage inhibition efficiency is 52%.

[MOBMIM][FAP] is acting as an anodic inhibitor in this case. It is possible that during anodic scan, the [FAP] anion complexes with the oxidation products of the stainless steel resulting in the formation of a cohesive layer on the surface of the substrate. Figure 11 shows the potentiodynamic polarization curves at varying concentrations of the ionic liquid. The Tafel plots of the anodic and cathodic scans are shown

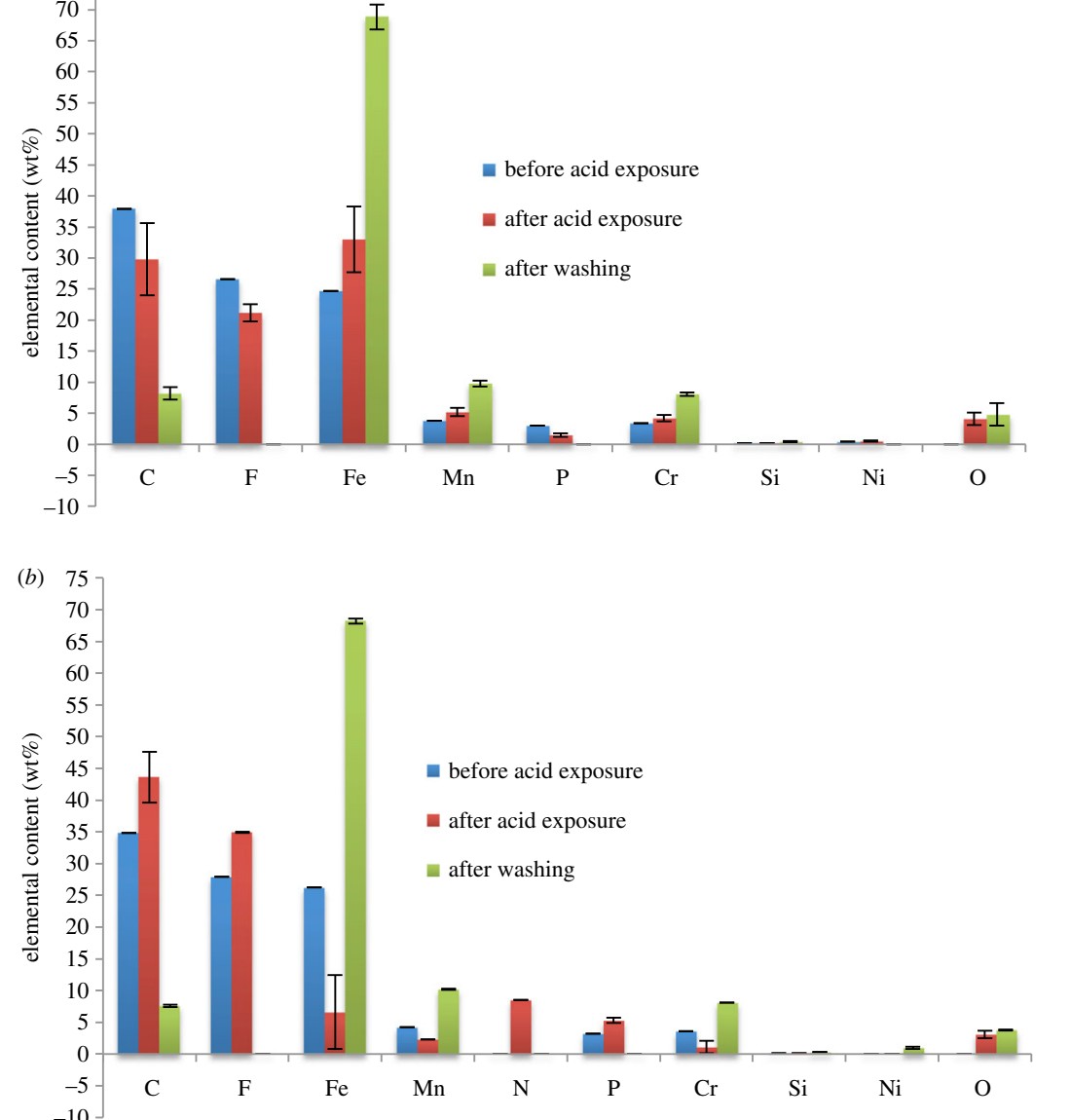

**Figure 10.** EDX results on the S20200 stainless steel substrate. (*a*) S20200 with [PMIM][FAP] and (*b*) S20200 with [MOBMIM][FAP].

**Table 3.** Ratio of %O and %Fe from the EDX results.

| substrate | $\left[\dfrac{\%; \text{ composition O}}{\%; \text{ composition Fe}}\right] \pm \delta x^{a}$ | | | |
|---|---|---|---|---|
| S30400 | without IL | | with IL | |
| | 0.07 ± 0.02 | | [PMIM][FAP] | 0.07 ± 0.01 |
| | | | [MOBMIM][FAP] | 0.02 ± 0.002 |
| S20200 | | | | |
| | 1.6 ± 0.28 | | [PMIM][FAP] | 0.07 ± 0.03 |
| | | | [MOBMIM][FAP] | 0.06 ± 0.002 |

[a]Calculated error propagation from the standard deviation of measured values.

in the electronic supplementary material. This was used to determine the corrosion exchange current density $J$ ($\mu A\ cm^{-2}$), and the overpotential $\eta$. The overpotential was then subtracted from the applied potential to obtain the corrosion potential ($E_{corr}$).

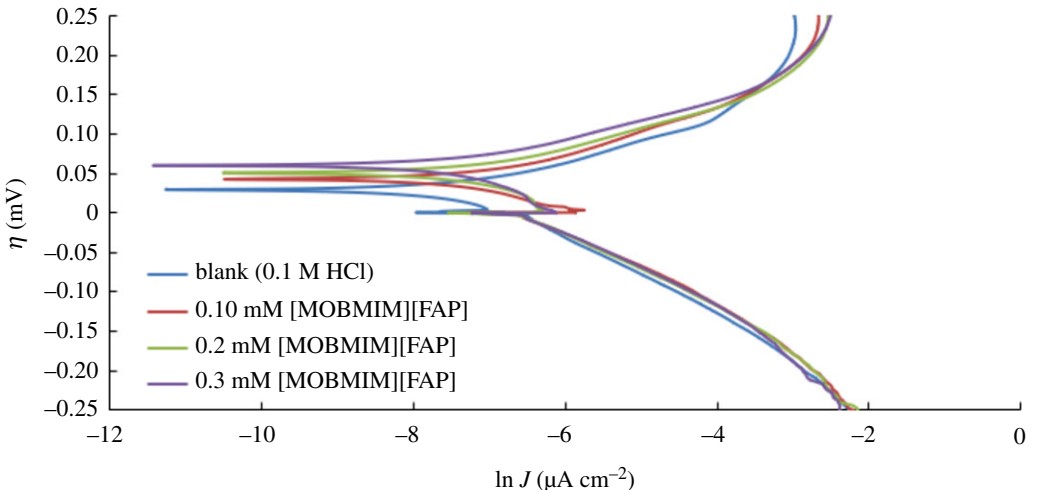

**Figure 11.** Potentiodynamic polarization curves of stainless steel in 1.0 M HCl with various concentrations of the inhibitor [MOBMIM][FAP] at room temperature.

**Table 4.** Polarization parameters of stainless steel in 1.0 M HCl at varying concentrations of inhibitor.

| Concentration (mM) | $\beta_{an}$ | $\beta_{cat}$ | $J_{corr}$ ($\mu A\ cm^{-2}$) | $-E_{corr}$ (mV versus Ag/AgCl) | % IE |
|---|---|---|---|---|---|
| blank | 0.029 | 0.050 | 764 | 365 | 0 |
| 0.10 | 0.023 | 0.046 | 580 | 372 | 24 |
| 0.20 | 0.021 | 0.046 | 459 | 372 | 40 |
| 0.30 | 0.019 | 0.044 | 370 | 380 | 52 |

## 4. Conclusion

Surface techniques SEM and EDX were used to investigate the corrosion inhibiting properties of FAP-based ionic liquids on stainless steel substrates S30400 and S20200. The resulting SEM images were able to show how these ionic liquids protected the stainless steel surfaces from corrosion. EDX results complemented the SEM results, where the calculated oxygen to iron percentage composition ratio showed that in the presence of the ionic liquids, corrosion was lessened, particularly on the S20200 surface. To ensure the inhibition capability of [MOBMIM][FAP] on S20200, which was quite promising based on the SEM–EDX results, potentiodynamic studies were performed. Electrochemical results also indicated the ability of [MOBMIM][FAP] to protect the stainless steel surface.

Measured CAs of the ionic liquids to the stainless steel substrates showed that the ionic liquids were able to wet the samples surfaces. A higher affinity of the [MOBMIM][FAP] ionic liquid on the S20200 substrate was observed, while [PMIM][FAP] on the S30400 substrate.

Future studies should include abrasion, plasma treatment or use of preliminary coating in the sample preparation of the substrates. Changing conditions like temperature, as well as employment of other techniques such as XPS will also provide additional information on the ability of these ionic liquids to inhibit corrosion on stainless steel. A more thorough study on the corrosion products and by-products in the presence of the ionic liquids will also be very helpful.

Data accessibility. The datasets supporting this article have been uploaded as part of the electronic supplementary material.

Authors' contributions. J.K.A.T. performed most of the experiments including sample preparation and characterization. He also wrote the first draft of the manuscript. I.S.M. as the principal investigator conceptualized the study, interpreted results as well as revised the manuscript for final submission. D.A.V.B., on the other hand, performed the potentiodynamic studies, and G.A.T. assisted in the interpretation of the electrochemical data. K.C.C.A. assisted J.K.A.T. in performing the SEM–EDX experiments, while A.P.G. and B.A.B., as the institute heads of ADMATEL helped in the analyses and data interpretation of the SEM–EDX results.

Competing interests. We have no competing interests.

Funding. The authors are thankful for the funding support provided by the Natural Sciences Research Institute (NSRI) (Project CHE-16) of the University of the Philippines-Diliman, the Advanced Device and Materials Testing Laboratory (ADMATEL)—ITDI in the Department of Science and Technology (DOST)—Philippines, and the Balik PhD Program of the University of the Philippines (FCM 2013-401, FCM 2012-372, FCM 2013-123).

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
