## [Reviewer comments · Royal Society Open Science]

Review History

RSOS-200580.R0 (Original submission)

Review form: Reviewer 1

Is the manuscript scientifically sound in its present form?

Yes

Are the interpretations and conclusions justified by the results?

Yes

Is the language acceptable?

Yes

Do you have any ethical concerns with this paper?

No

Have you any concerns about statistical analyses in this paper?

No

Recommendation?

Accept with minor revision (please list in comments)

Comments to the Author(s)

In the discussion of Figure 9, it should be stated what the regions a and b refer to.

In Table 3, there is no need to abbreviate what are presumably "without" and "with".

Review form: Reviewer 2**Is the manuscript scientifically sound in its present form?**

Yes

Are the interpretations and conclusions justified by the results?

Yes

Is the language acceptable?

Yes

Do you have any ethical concerns with this paper?

No

Have you any concerns about statistical analyses in this paper?

No

Recommendation?

Major revision is needed (please make suggestions in comments)

Comments to the Author(s)

The manuscript describes a study of FAP-based ionic liquids (ILs) as anti-corrosion reagents for stainless steel. The FAP anion is known to be stable against acids compared with BF₄ and PF₆ anions and therefore was used as the IL anion in the present study. Because the results are interesting I do not deny the publication of the manuscript to RSOS. The followings are major amendments before the publication.

Amide anions such as TFSA and BETI are also hydrophobic and acid-stable. Another good point is the cost; they are much cheaper than FAP. The authors should comment on why they used FAP in Introduction.

I cannot understand the discussion of Figs. 6 and 7 using "excess positive charge" and "excess negative charge" (p.5 4th paragraph). Is the "charge" located on the steel surface, the IL surface, or the steel/IL interface? Why and how did the positive charge generate at first? Why does the negative charge make IL spread on the surface? Also, they attributed the formation of 4 μm size IL globules to micelle formation, but 4 μm is too big (three orders of magnitude bigger) for micelles.

I also do not understand "with EDX exposure and minus EDX exposure" in Fig.7 caption. Does "EDX exposure" mean electron beam irradiation? What is "minus EDX exposure"? They should clearly explain what they did.

Miscellaneous

Please define PREN (Table 2)

work has -> work, has (p.2, l.26)

where -> were (p.2, l.40)

by the group -> by our group (p.2, l.52)

filliform -> filiform (p.4, l.19)
 approximately 3.85 um -> approximately 4 um (p.5, l.29)
 however were -> however, was (p.5, l.41)
 there was not -> there were not (p.5, l.42)
 are be -> are (p.5, l.59)
 $\ln J$ -> $\ln(J/A \text{ cm}^{-2})$? (Fig.11 caption)

Decision letter (RSOS-200580.R0)

Dear Dr Martinez:

Title: Exploring the Corrosion Inhibition Capability of
 FAP-based Ionic Liquids on Stainless Steel
 Manuscript ID: RSOS-200580

Thank you for submitting the above manuscript to Royal Society Open Science. On behalf of the Editors and the Royal Society of Chemistry, I am pleased to inform you that your manuscript will be accepted for publication in Royal Society Open Science subject to minor revision in accordance with the referee suggestions. Please find the reviewers' comments at the end of this email.

The reviewers and handling editors have recommended publication, but also suggest some minor revisions to your manuscript. Therefore, I invite you to respond to the comments and revise your manuscript.

Because the schedule for publication is very tight, it is a condition of publication that you submit the revised version of your manuscript before 22-May-2020. Please note that the revision deadline will expire at 00.00am on this date. If you do not think you will be able to meet this date please let me know immediately.

- 1) A text file of the manuscript (tex, txt, rtf, docx or doc), references, tables (including captions) and figure captions. Do not upload a PDF as your "Main Document".
- 2) A separate electronic file of each figure (EPS or print-quality PDF preferred (either format should be produced directly from original creation package), or original software format)

- 3) Included a 100 word media summary of your paper when requested at submission. Please ensure you have entered correct contact details (email, institution and telephone) in your user account
- 4) Included the raw data to support the claims made in your paper. You can either include your data as electronic supplementary material or upload to a repository and include the relevant doi within your manuscript
- 5) All supplementary materials accompanying an accepted article will be treated as in their final form. Note that the Royal Society will neither edit nor typeset supplementary material and it will be hosted as provided. Please ensure that the supplementary material includes the paper details where possible (authors, article title, journal name).

Kind regards,
Dr Laura Smith
Publishing Editor, Journals

On behalf of the Subject Editor Professor Anthony Stace and the Associate Editor Dr Darren Walsh.

RSC Associate Editor:
Comments to the Author:
(There are no comments.)

RSC Subject Editor:
Comments to the Author:
(There are no comments.)

Reviewer comments to Author:
Reviewer: 1

Comments to the Author(s)
In the discussion of Figure 9, it should be stated what the regions a and b refer to.

In Table 3, there is no need to abbreviate what are presumably "without" and "with".

Reviewer: 2

Comments to the Author(s)

The manuscript describes a study of FAP-based ionic liquids (ILs) as anti-corrosion reagents for stainless steel. The FAP anion is known to be stable against acids compared with BF₄ and PF₆ anions and therefore was used as the IL anion in the present study. Because the results are interesting I do not deny the publication of the manuscript to RSOS. The followings are major amendments before the publication.

Amide anions such as TFSA and BETI are also hydrophobic and acid-stable. Another good point is the cost; they are much cheaper than FAP. The authors should comment on why they used FAP in Introduction.

I cannot understand the discussion of Figs. 6 and 7 using “excess positive charge” and “excess negative charge” (p.5 4th paragraph). Is the “charge” located on the steel surface, the IL surface, or the steel/IL interface? Why and how did the positive charge generate at first? Why does the negative charge make IL spread on the surface? Also, they attributed the formation of 4 μm size IL globules to micelle formation, but 4 μm is too big (three orders of magnitude bigger) for micelles.

I also do not understand “with EDX exposure and minus EDX exposure” in Fig.7 caption. Does “EDX exposure” mean electron beam irradiation? What is “minus EDX exposure”? They should clearly explain what they did.

Miscellaneous

Please define PREN (Table 2)

work has -> work, has (p.2, l.26)

where -> were (p.2, l.40)

by the group -> by our group (p.2, l.52)

filliform -> filiform (p.4, l.19)

approximately 3.85 μm -> approximately 4 μm (p.5, l.29)

however were -> however, was (p.5, l.41)

there was not -> there were not (p.5, l.42)

are be -> are (p.5, l.59)

lnJ -> ln(J/A cm⁻²) ? (Fig.11 caption)

Author's Response to Decision Letter for (RSOS-200580.R0)

See Appendix A.

Decision letter (RSOS-200580.R1)

Dear Dr Martinez:

Title: Exploring the Corrosion Inhibition Capability of FAP-based Ionic Liquids on Stainless Steel

Manuscript ID: RSOS-200580.R1

It is a pleasure to accept your manuscript in its current form for publication in Royal Society Open Science. The chemistry content of Royal Society Open Science is published in collaboration with the Royal Society of Chemistry.

On behalf of the Subject Editor Professor Anthony Stace and the Associate Editor Dr Darren Walsh.

RSC Associate Editor
Comments to the Author:
(There are no comments.)

Reviewer(s)' Comments to Author:

Appendix A

Exploring the Corrosion Inhibition Capability of [FAP]- based Ionic Liquids on Stainless Steel

Julius Kim A. Tiongson^a, Kim Christopher C. Aganda^b,

Albert P. Guevara^b, Dwight Angelo V. Bruzon^c, Blessie A. Basilia^b, Giovanni A. Tapang^c, and Imee Su Martinez^{*, a, d}

^a Natural Sciences Research Institute, University of the Philippines Diliman, Quezon City, Philippines, 1101

^b Advanced Device and Materials Testing Laboratory, Department of Science and Technology Compound, Gen. Santos Ave., Bicutan, Taguig City, Philippines, 1631

^c National Institute of Physics University of the Philippines Diliman, Quezon City, Philippines, 1101

^d Institute of Chemistry, University of the Philippines Diliman, Quezon City, Philippines, 1101

*Corresponding author: ismartinez@up.edu.ph

Answers to Comments:

Reviewer comments to Author:

Reviewer: 1

Comments to the Author(s)

In the discussion of Figure 9, it should be stated what the regions a and b refer to.

Answer: Revision performed. Regions a and b are now mentioned in discussion, pages 5-6.

In Table 3, there is no need to abbreviate what are presumably "without" and "with".

Answer: Revision done. The words with and without are spelled out, and no longer abbreviated.

Reviewer: 2

Comments to the Author(s)

The manuscript describes a study of FAP-based ionic liquids (ILs) as anti-corrosion reagents for stainless steel. The FAP anion is known to be stable against acids compared with BF₄ and PF₆ anions and therefore was used as the IL anion in the present study. Because the results are interesting I do not deny the publication of the manuscript to RSOS. The followings are major amendments before the publication.

Amide anions such as TFSA and BETI are also hydrophobic and acid-stable. Another good point is the cost; they are much cheaper than FAP. The authors should comment on why they used FAP in Introduction.

Answer: It is widely accepted that depending on cation-anion combinations, properties of ionic liquids may vary. Our interest in testing imidazolium FAP ionic liquids as corrosion inhibition materials, stem from the results of our previous work where determined properties of these ILs such as electrochemical stability, hydrophobicity, and thermal stability were found to be quite promising for this specific application.¹ In fact these properties were found to be comparable to that of Tf₂N/TFSA ionic liquids given of course similar cations. For the property electrochemical stability, the FAP ionic liquids were found to possess even wider potential windows keeping in mind differences in electrodes used. This is now added and referred to in the introduction part to show why FAP is being used.

Since TFSA is known to be more electrochemically stable than BETI, BETI is no longer included in the manuscript.² Granted that FAP ionic liquids are more expensive, the fact remains that their properties are quite unknown, and that it will be interesting to determine how well they work for specific applications. Nominal cost does not necessarily translate to economic efficiency given high efficacy of a material. Also, the difficulty in synthesizing FAP ionic liquids, which previously can only be performed through electrofluorination led to its high cost, which hopefully can now be addressed through anion-exchange chromatography. Perhaps another paper comparing these FAP ionic liquids to the sulfonyl types in terms of corrosion inhibition properties will be an interesting worthwhile endeavor.

References:

¹Tiongson, J. K., Bruzon, D. A., Tapang, G. A., Martinez, I. S., J. Chem. Eng. Data 2018, 63, 5, 1135-1145.

²De Vos, N., Maton, C., Stevens, C. V., ChemElectroChem 2014, 1, 1258 – 1270.

I cannot understand the discussion of Figs. 6 and 7 using “excess positive charge” and “excess negative charge” (p.5 4th paragraph). Is the “charge” located on the steel surface, the IL surface, or the steel/IL interface? Why and how did the positive charge generate at first? Why does the negative charge make IL spread on the surface? Also, they attributed the formation of 4 μm size IL globules to micelle formation, but 4 μm is too big (three orders of magnitude bigger) for micelles.

Answer: S30400 stainless steel having more chromium oxide and hydroxide on its surface is known to have an isoelectric point of around pH=4.3. When it is subjected to a lower pH, the surface is covered with $-\text{OH}_2^+$. On the other hand, at higher pH the hydroxyl groups is converted to $-\text{OH}^-$. In this study, the stainless steel with the ionic liquid is subjected to aqueous acid, leading to the predominance of $-\text{OH}_2^+$ on its surface. Since the ionic liquid is hydrophobic, it will probably tend to withdraw into itself away from the aqueous acid. Since ionic liquid ions also tend to layer depending on the charge of the surface they are in contact with, their anions will probably be situated closer to the stainless surface while the cations will be situated away from the positive surface of the stainless steel. Both of these things possibly led to the observed globule formation in the experiment. When the SEM image was taken, subjecting the overall system to a stream of electrons, the electrons must have been readily attracted to the more positive charge of the stainless steel surface as opposed to the more neutral ionic liquid (both ions present), causing an excess negative charge on the stainless steel surface. This must have made the cations of the ionic liquids move towards the exposed portion of the stainless steel after some time, causing them to spread. However, as mentioned in the manuscript, additional studies have to be performed to understand the phenomenon better.

Revisions were made on the discussion to make the explanation more clear, and to also add the term aggregation as opposed to just micelle formation.

I also do not understand “with EDX exposure and minus EDX exposure” in Fig.7 caption. Does “EDX exposure” mean electron beam irradiation? What is “minus EDX exposure”? They should clearly explain what they did.

Answer: Revisions done. “With EDX exposure” was changed to “after taking EDX data”, and “minus EDX exposure” changed to “prior to taking EDX data”.

Basically, the process entailed taking the SEM image (Fig.7b) of the stainless substrate with the [MOBMIM][FAP] ionic liquid after acid exposure. We then took the EDX data from the same substrate, and then subsequently took another SEM image (Fig.7a).

Miscellaneous

Please define PREN (Table 2)

work has -> work, has (p.2, l.26)

where -> were (p.2, l.40)

by the group -> by our group (p.2, l.52)

filliform -> filiform (p.4, l.19)

approximately 3.85 um -> approximately 4 um (p.5, l.29)

however were -> however, was (p.5, l.41)

there was not -> there were not (p.5, l.42)

are be -> are (p.5, l.59)

$\ln J$ -> $\ln(J/A \text{ cm}^{-2})$? (Fig.11 caption)

Answer: Revisions all done